# Regulation of cilia abundance in multiciliated cells

Rashmi Nanjundappa[1], Dong Kong[2], Kyuhwan Shim[1], Tim Stearns[3], Steven L Brody[4], Jadranka Loncarek[2], Moe R Mahjoub[1,5]*

[1]Nephrology Division, Department of Medicine, Washington University, St Louis, United States; [2]Center for Cancer Research, National Cancer Institute, Frederick, United States; [3]Department of Biology, Stanford University, Stanford, United States; [4]Pulmonary Division, Department of Medicine, Washington University, St Louis, United States; [5]Department of Cell Biology and Physiology, Washington University, St Louis, United States

**Abstract** Multiciliated cells (MCC) contain hundreds of motile cilia used to propel fluid over their surface. To template these cilia, each MCC produces between 100-600 centrioles by a process termed centriole amplification. Yet, how MCC regulate the precise number of centrioles and cilia remains unknown. Airway progenitor cells contain two parental centrioles (PC) and form structures called deuterosomes that nucleate centrioles during amplification. Using an ex vivo airway culture model, we show that ablation of PC does not perturb deuterosome formation and centriole amplification. In contrast, loss of PC caused an increase in deuterosome and centriole abundance, highlighting the presence of a compensatory mechanism. Quantification of centriole abundance in vitro and in vivo identified a linear relationship between surface area and centriole number. By manipulating cell size, we discovered that centriole number scales with surface area. Our results demonstrate that a cell-intrinsic surface area-dependent mechanism controls centriole and cilia abundance in multiciliated cells.
DOI: https://doi.org/10.7554/eLife.44039.001

*For correspondence:
mmahjoub@wustl.edu

Competing interests: The authors declare that no competing interests exist.

## Introduction

In mammals, multiciliated cells (MCC) are terminally differentiated epithelia that line the respiratory tract, brain ventricles and segments of the female and male reproductive organs (*Brooks and Wallingford, 2014*; *Spassky and Meunier, 2017*). Each MCC contains hundreds of motile cilia, microtubule-based organelles that generate the motive force to move fluid over the surface of the cell. In the airway, motile cilia of MCC beat directionally to propel mucus and inhaled contaminants out of the lungs. Ciliary motility of ependymal MCC lining the brain ventricles is key for movement of cerebrospinal fluid through the central nervous system, while cilia on MCC in the reproductive tracts are important for ovum and sperm transport (*Brooks and Wallingford, 2014*; *Spassky and Meunier, 2017*). Thus, the proper assembly of hundreds of motile cilia is critical for the functions of MCC, and the overall health of the associated tissues. Mutations that result in reduced cilia abundance cause human diseases due to aberrant cilia-based fluid flow (*Amirav et al., 2016*; *Boon et al., 2014*; *Núñez-Ollé et al., 2017*; *Wallmeier et al., 2014*), suggesting that establishment of the correct number of cilia per cell is important for function. MCC in different tissues display significant variability in motile cilia number, ranging from 30 to 600 cilia per cell (*Brooks and Wallingford, 2014*; *Spassky and Meunier, 2017*). However, a major unresolved question is how each cell regulates the precise number of its motile cilia during differentiation.

To template these cilia, each MCC undergoes a process termed centriole amplification to produce hundreds of centrioles, barrel-shaped microtubule structures that form the base upon which

cilia are assembled. In the airway, proliferating progenitors (called basal cells) initially contain a single centrosome with two parental centrioles (PC), which they maintain via the canonical centriole duplication mechanism (*Figure 1a*; *Nigg and Holland, 2018*; *Sorokin, 1968*; *Vladar and Stearns, 2007*). During MCC differentiation, the post-mitotic basal cells undergo massive centriole production that was thought to occur through two parallel pathways: a parental centriole-dependent pathway whereby the original PC template the assembly of multiple procentrioles simultaneously, and an acentriolar or 'de novo' pathway mediated by dozens of deuterosomes, spheroidal electron-dense structures that are capable of producing multiple procentrioles (*Figure 1a*; *Kalnins and Porter, 1969*; *Sorokin, 1968*; *Steinman, 1968*). Deuterosomes are transient structures that are absent in proliferating progenitor cells, are formed during early centriole amplification, and mostly disappear after centriologenesis is complete (*Klos Dehring et al., 2013*; *Zhao et al., 2013*). Although both PC-dependent and deuterosome-dependent pathways have been known for decades, the molecular mechanisms that govern the two pathways and their relative contributions to the total complement of centrioles has remained enigmatic.

Most of the key proteins involved in canonical centriole duplication play an essential role in both PC- and deuterosome-dependent centriole amplification pathways in MCC (*Klos Dehring et al., 2013*; *Mahjoub et al., 2010*; *Tang, 2013*; *Vladar and Stearns, 2007*; *Zhao et al., 2013*). This indicates that centriole duplication in cycling cells and centriole amplification in MCC share common and conserved molecular mechanisms despite their morphological differences. However, recent studies have characterized proteins that are uniquely associated with deuterosomes. For example, Deup1 was recently identified as a deuterosome-specific protein that is essential for the formation of deuterosomes and, subsequently, centriole amplification (*Zhao et al., 2013*). Similarly, a number of deuterosome-enriched factors have since been shown to play key roles in deuterosome biogenesis and function (*Klos Dehring et al., 2013*; *Mori et al., 2017*; *Revinski et al., 2018*). These data support the theory that the PC-dependent and deuterosome-dependent centriole amplification pathways act in parallel, but use unique molecular components to produce all of the centrioles in a cell. It has been estimated that the deuterosome-dependent pathway contributes the majority (80–90%) of the total centrioles assembled during amplification, while the PC-dependent pathway contributes roughly 10–20% towards the final number (*Al Jord et al., 2014*; *Sorokin, 1968*; *Spassky and Meunier, 2017*; *Zhao et al., 2013*). Intriguingly, a recent study found that deuterosomes themselves are nucleated from the original PC, indicating that these two pathways may not be independent of each other after all (*Al Jord et al., 2014*). If true, this would suggest that all centrioles produced in MCC are ultimately dependent on the PC. Yet, a number of important questions have remained unanswered: are parental centrioles necessary for deuterosome biogenesis? What is the actual contribution of the PC to final centriole number? How do cells that start with exactly two parental centrioles generate such a variable number during differentiation?

In this study, we sought to determine the mechanisms by which airway MCC establish centriole number, using an ex vivo airway culture system (*Vladar and Brody, 2013*; *You and Brody, 2013*; *You et al., 2002*). We began by testing the hypothesis that parental centrioles are essential for deuterosome formation. To ablate PC during proliferation, we used a small molecule inhibitor and shRNA-mediated depletion of Plk4, the major kinase involved in centriole duplication (*Habedanck et al., 2005*; *Nigg and Holland, 2018*). Surprisingly, loss of parental centrioles did not affect deuterosome formation, indicating that PC are dispensable for deuterosome biogenesis. Moreover, the loss of PC and Plk4 function did not disrupt centriole amplification nor decrease the total number of centrioles per cell. Thus, establishment of centriole abundance is regulated by a different cellular property. Airway MCC vary in size and surface area, and display a wide range in centriole number. Quantification of centriole abundance in vitro and in vivo highlighted a direct relationship between cell-surface area and centriole number. To determine if cell size was a major determinant of centriole abundance, we cultured basal cells on an extracellular matrix of increasing density to manipulate size. We found that centriole number scales with increasing surface area, but not volume, of multiciliated cells. Together, our results point to a cell-intrinsic, surface area-dependent mechanism that establishes centriole-cilia abundance in airway multiciliated cells.

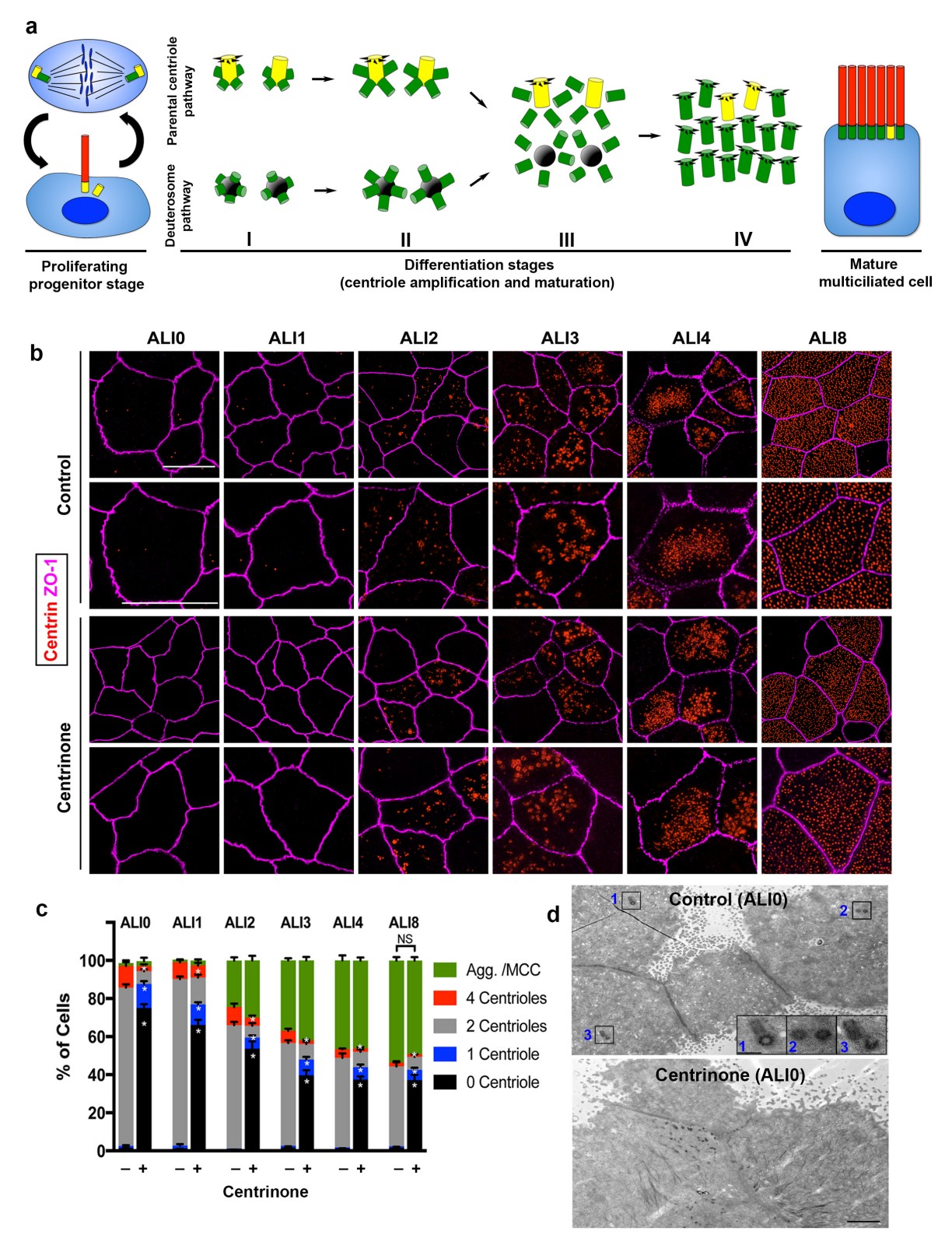

**Figure 1.** Parental centrioles are dispensable for centriole amplification. (**a**) Schematic of the proliferation and differentiation steps of multiciliated respiratory epithelia. The key stages of centriole amplification from parental centrioles and deuterosomes are depicted. (**b**) Immunofluorescence (3D-SIM) images of control and Centrinone-treated cells. Samples were fixed on the indicated days of air-liquid interface (ALI) culture and stained for centrioles (centrin) and apical cell-cell junctions (ZO-1). Upper panels show lower magnification images, and lower panels highlight a single cell. Scale

*Figure 1 continued on next page*

*Figure 1 continued*

bar = 10 µm. (**c**) Quantification of centriole number, and the fraction of the population undergoing centriole amplification, in control and Centrinone-treated MTEC. Ablation of parental centrioles did not affect the initiation of centriolar aggregate (Agg.) formation, the timing of centriole amplification, or the fraction of mature multiciliated cells (MCC) in the population. Results are averages of three independent experiments. More than 3,000 cells were counted per sample for each time point. *p<0.05. (**d**) Transmission electron microscopy (TEM) of the apical surface of MTEC at ALI0. Parental centrioles were easily found at the apical surface of control cells (20/20 cells examined; insets highlight three sets of PC), but were missing in the majority of Centrinone-treated samples (0 parental centrioles in 17/19 cells). Scale bars = 2 µm (large panels) and 400 nm (insets).
DOI: https://doi.org/10.7554/eLife.44039.002
The following video is available for figure 1:
**Figure 1—video 1.** Centrinone-mediated inhibition of Plk4 results in loss of parental centrioles at ALI0.
DOI: https://doi.org/10.7554/eLife.44039.003

## Results

### Parental centrioles are dispensable for centriole amplification

We previously established a mouse tracheal epithelial cell (MTEC) culture system to study centriologenesis and ciliary assembly in MCC (*Mahjoub et al., 2010*; *Silva et al., 2016*; *You and Brody, 2013*; *You et al., 2002*). This airway model entails seeding freshly isolated mouse tracheal basal cells onto a porous filter suspended in medium, and allowing the progenitors to proliferate into a confluent, polarized layer. Cells are induced to differentiate into MCC and other cell types found in the airway (for example, secretory cells) by exposure to an air–liquid interface (ALI). During proliferation, basal cells contain two centrosomal centrioles that are maintained via the canonical centriole duplication mechanism (*Figure 1a*). During post-mitotic MCC differentiation, the centriologenesis program proceeds in distinct stages from ALI days 0 to 12: centriolar protein aggregates and deuterosomes form near the PC, and procentrioles are nucleated from both PC and deuterosomes (stage I, ALI1-2); procentrioles continue to grow and elongate from PC and deuterosomes (stage II, ALI3-4); centrioles disengage from PC and deuterosomes, start to mature into basal bodies and migrate towards the apical membrane (stage III, ALI4-8); fully mature, apically docked basal bodies template ciliary assembly (stage IV, ALI8-12). By ALI12, most cells have completed differentiation into fully mature MCC (*Figure 1a*).

It was recently proposed that deuterosomes are nucleated from the original PC, suggesting that the de novo centriole assembly pathway is dependent on PC (*Al Jord et al., 2014*). To test this theory, we ablated PC in basal cells using Centrinone (*Wong et al., 2015*), a selective inhibitor of the kinase Plk4 known as the 'master regulator' of centriole assembly (*Arquint and Nigg, 2016*; *Bettencourt-Dias et al., 2005*; *Habedanck et al., 2005*; *Pearson and Winey, 2010*; *Sillibourne and Bornens, 2010*). Centrinone blocks centriole duplication, resulting in cells that lack centrioles following a few rounds of cell division (*Wong et al., 2015*). Basal cells were cultured with or without 1 µM Centrinone, roughly 3–8 times higher concentrations than needed to suppress centriole formation in most cells (*Wong et al., 2015*), and the compound was maintained in the medium throughout differentiation (described in Materials and methods). Samples were fixed at various stages of differentiation, immunostained for centrioles and analyzed using either confocal or superresolution structured illumination microscopy (3D-SIM). Control cells at ALI0 predominantly contain the original two PC (*Figure 1b–c*). Upon exposure to ALI, cells proceeded through the various stages of centriole amplification, resulting in a population of cells comprised of roughly 60% MCC and 40% non-MCC as expected (*Figure 1c*). Growth of basal cells in the presence of 1 µM Centrinone resulted in >80% of cells lacking PC at ALI0 (*Figure 1b–c*). Remarkably, the formation of centriolar aggregates and subsequent centriole amplification proceeded normally in the absence of PC. Moreover, there was no significant difference in the timing of centriologenesis, or the fraction of MCC in the population (*Figure 1b–c*). To ensure that Centrinone-treated cells were indeed lacking PC prior to differentiation, we analyzed cells using transmission electron microscopy (TEM). Cells at ALI0 were serially sectioned beginning at the apical surface and down to the basal membrane. Whereas PC were evident in control cells, there was >80% decrease in cells containing PC upon Centrinone treatment (*Figure 1d* and *Figure 1—video 1*), consistent with the immunofluorescence analysis.

The abundance of Plk4 protein is regulated by autophosphorylation-induced degradation (*Holland et al., 2010*). Centrinone-mediated inhibition of Plk4 kinase activity blocks this

degradation, resulting in elevated (yet inactive) Plk4 protein levels (*Wong et al., 2015*). In control cells, Plk4 expression increased during early stages of centriole amplification, peaking at ALI3 and subsequently decreasing upon centriole maturation at ALI8 (*Figure 2a–b*). Centrinone-treated cells showed accumulation of Plk4, which was already evident in a higher proportion of cells at ALI0-1 (*Figure 2a*). Unlike control cells, the high Plk4-expressing fraction persisted at ALI8 and beyond (*Figure 2a–b*), indicating that Plk4 kinase activity was indeed blocked. Moreover, the non-multiciliated cells in the population did not reform/regain their PC (*Figure 1c*), further indicative of Centrinone's inhibitory activity in our assay. Finally, we examined whether loss of PC attenuated the various stages of centriole assembly. Control and Centrinone-treated cells were immunostained for markers of procentriole initiation (STIL, Cep152, Sas6, Cep135 [*Cizmecioglu et al., 2010*; *Dammermann et al., 2004*; *Hatch et al., 2010*; *Kleylein-Sohn et al., 2007*; *Tang et al., 2011*]), centriole growth and elongation (Cep120, centrin, γ-tubulin [*Betleja et al., 2018*; *Comartin et al., 2013*; *Mahjoub et al., 2010*; *Paoletti et al., 1996*; *Stearns et al., 1991*; *Xie et al., 2007*]), centriole maturation (Cep164 [*Graser et al., 2007*]), and ciliogenesis (acetylated-tubulin). Loss of PC did not affect the overall timing of centriole amplification stages, or the fraction of MCC in the population (*Figure 2c–g* and *Figure 2—figure supplements 1–2*). Centriole maturation and ciliogenesis were also unaffected upon loss of PC and Plk4-inhibition (*Figure 2—figure supplement 3*). Collectively, these results indicate that parental centrioles and Plk4 kinase activity are not necessary for centriole amplification and ciliogenesis in airway MCC.

## Parental centrioles are dispensable for deuterosome formation

Next, we tested whether loss of PC altered the expression pattern or number of deuterosomes in MCC. Cells at ALI3 were immunostained for Deup1, a core component of deuterosomes essential for their formation (*Zhao et al., 2013*). In control cells, deuterosome formation begins ~ALI1 and peaks at ALI3, and they are subsequently lost by ALI8 (*Figure 3a–b*). Surprisingly, Deup1 expression was grossly unchanged in cells lacking PC (*Figure 3a–b*). However, Deup1 foci were evident in a higher proportion of cells at ALI1, suggesting that deuterosome formation may start sooner in PC-less cells. Since MCC are variable in size and centriole number (discussed in more detail in the subsequent section), we measured deuterosome number and cell surface area (using ZO-1 to outline apical cell-cell junctions), then compared deuterosome abundance in cells of similar size (*Figure 3c*). Centrinone-treated cells lacking PC displayed a slight but significant increase in average deuterosome number per cell (*Figure 3d*). Serial-section TEM analysis of cells at ALI3 confirmed the presence of deuterosome structures that were nucleating procentrioles in PC-less cells (*Figure 3e*). These deuterosomes appeared consistent in size and morphology with control cells. Moreover, deuterosomes in PC-less cells nucleated roughly similar amounts of procentrioles each as compared to control cells (*Figure 3—figure supplement 1*), with a slight decrease in average procentriole number per deuterosome.

The deuterosome-dependent pathway is thought to contribute the majority (80–90%) of centrioles assembled during amplification, whereas the PC-dependent pathway contributes roughly 10–20% towards the final complement (*Al Jord et al., 2014*; *Sorokin, 1968*; *Spassky and Meunier, 2017*; *Zhao et al., 2013*). Therefore, we reasoned that loss of PC might result in a reduction in the final complement of centrioles in mature MCC. Quantification of centriole number in control cells at ALI12 showed a range from 100 to 600, and correlated with increasing surface area (*Figure 4a–b*). Loss of PC in Centrinone-treated cells did not disrupt this relationship (*Figure 4c*). Importantly, the average number of centrioles per mature MCC did not decrease; in contrast we noted a slight but significant increase in centriole abundance and density (*Figure 4d–e*). All together, these results indicate that loss of PC does not inhibit deuterosome biogenesis nor result in decreased centriole abundance in MCC.

## Modulating Plk4 expression delays centriole assembly but does not affect number

We were surprised to discover that inhibiting Plk4 activity had no detrimental effect on centriole amplification, as this kinase is known to play a critical role in centriole assembly in the majority of cell types examined in mammals. Plk4 expression becomes elevated during centriole amplification in airway MCC (*Hoh et al., 2012*), where it localizes to both PC and deuterosomes (*Zhao et al., 2013*),

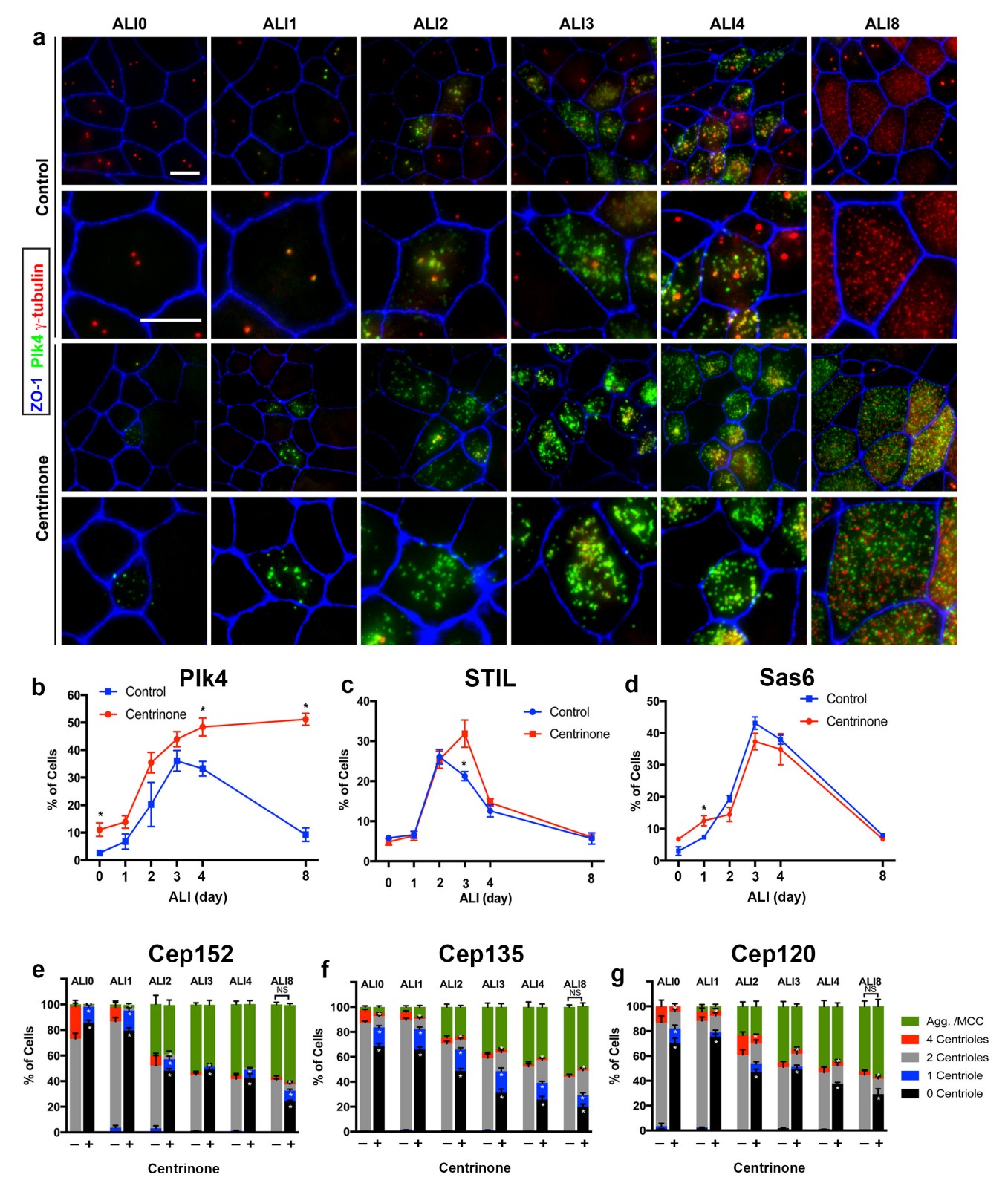

**Figure 2.** Loss of parental centrioles does not affect dynamics of centriole amplification. (a) Immunofluorescence images of control and Centrinone-treated cells. Samples were fixed on the indicated days and stained for Plk4, centrioles (γ-tubulin) and apical cell-cell junctions (ZO-1). Upper panels show lower magnification images, and lower panels highlight a single cell. Inhibition of Plk4 kinase activity results in accumulation of Plk4 protein at ALI 0–1, and the protein is still evident at stages (e.g. ALI 8) when it is normally lost. Scale bar = 10 µm. (b) Quantification of the fraction of Plk4-expressing

*Figure 2 continued on next page*

*Figure 2 continued*

cells during differentiation. Results are averages of two independent experiments. More than 1,500 cells were counted per sample for each time point. *p<0.05. (**c**) Quantification of the fraction of STIL-expressing cells during differentiation. STIL expression is mostly unchanged upon inhibition of Plk4, with a slight increase in STIL-expressing cells evident at ALI three in Centrinone-treated samples. Results are averages of two independent experiments. More than 1000 cells were counted per sample for each time point. *p<0.05. (**d**) Quantification of the fraction of Sas6-expressing cells during differentiation. A slight increase in Sas6-expressing cells is evident at ALI 0–1 in Centrinone-treated samples, however the overall pattern of expression is not affected. Results are averages of two independent experiments. More than 1000 cells were counted per sample for each time point. *p<0.05. (**e–f**) Quantification of centriole number and the fraction of the population undergoing centriole amplification. Cells at the indicated time points were stained for markers of procentriole initiation and growth. Ablation of parental centrioles did not affect the formation of centriolar aggregates (Agg.), the timing of centriole amplification and growth, or the fraction of mature multiciliated cells (MCC) in the population. Results are averages of two independent experiments. More than 2000 cells were counted per sample for each time point. *p<0.05.

DOI: https://doi.org/10.7554/eLife.44039.004

The following figure supplements are available for figure 2:

**Figure supplement 1.** – Analysis of procentriole assembly in control and Centrinone-treated MTEC.
DOI: https://doi.org/10.7554/eLife.44039.005
**Figure supplement 2.** – Analysis of procentriole growth in control and Centrinone-treated MTEC.
DOI: https://doi.org/10.7554/eLife.44039.006
**Figure supplement 3.** – Analysis of centriole maturation and ciliogenesis in control and Centrinone-treated MTEC.
DOI: https://doi.org/10.7554/eLife.44039.007

thus it has been generally assumed to play an important role in centriole amplification. However, our Centrinone treatments suggest that although Plk4 kinase activity is necessary for centriole duplication during the proliferation phase of basal cells, it is not essential for post-mitotic centriole amplification (*Figures 1–3*). One possibility is that Plk4 acts in a kinase-independent manner to control centriole amplification, for example as a scaffold to recruit other components of the centriole assembly pathway. To test this, we depleted the protein in basal cells using shRNA-expressing lentivirus. Depletion of Plk4 during the proliferation phase resulted in near complete loss of PC by ALI0 (*Figure 5a–b*), similar to the Centrinone experiments. Examination of cells at ALI3 showed that cells lacking Plk4 still initiate centriologenesis similar to control cells (*Figure 5a,c and d*), and form more deuterosomes on average (*Figure 5a and e*), consistent with the Centrinone-mediated experiments. Moreover, quantification of the fraction of MCC at ALI12 showed no overall difference between control and Plk4-depleted cells. However, the process of centriole amplification appeared delayed, as roughly 50% of Plk4-depleted cells at ALI12 were in Stages I-II or III-IV instead of being fully mature MCC (*Figure 5a and f*). We interpret these results to suggest that loss of Plk4 delays the process, but does not impact overall centriole assembly. Indeed, culturing these Plk4-depleted cells for an additional 9 days caused an increase in the fraction of mature MCC (*Figure 5a and g*). Importantly, once cells completed maturation we did not observe a significant decrease in centriole number per cell (*Figure 5h*). Finally, we used a combination of shRNA-mediated depletion and Centrinone-based inhibition of Plk4 to ensure that any residual Plk4 activity/function was blocked. We found that the dynamics of centriole biogenesis were almost identical to shRNA-mediated depletion (*Figure 5—figure supplement 1*), with no additive effects. Collectively, these results indicate that loss of Plk4 delays the centriole assembly program but does not decrease centriole abundance.

Loss of PC and Plk4 function did not reduce centriole abundance in mature MCC, suggesting that establishment of centriole number is not dependent on them. To further test this hypothesis, we induced the formation of excess PC in basal cells by overexpressing Plk4, which drives nucleation of ectopic centrioles in a diversity of cells and organisms (*Habedanck et al., 2005*; *Sillibourne and Bornens, 2010*). To overexpress Plk4 in a temporally controlled manner, we used a recently developed mouse model (Tg::mChPlk4) whereby mCherry-tagged Plk4 is regulated by tamoxifen-induced Cre-recombinase expression (*Dionne et al., 2018*; *Marthiens et al., 2013*). To conditionally express Plk4 in MTEC, we bred Tg::mChPlk4 mice to a Tg::Rosa26-Cre^ERT2 (*Ventura et al., 2007*) strain that globally expresses Cre upon tamoxifen addition. Basal cells were harvested from trachea of Tg::mChPlk4/Rosa26-Cre^ERT2 mice, and MTEC cultures established in the presence or absence of tamoxifen. Constitutive expression of mChPlk4 during the proliferation stage resulted in cells that contained more than the normal complement of two PC at ALI0 (*Figure 5i*). However, mature MCC containing excess PC and Plk4 protein formed a similar number of centrioles when compared to

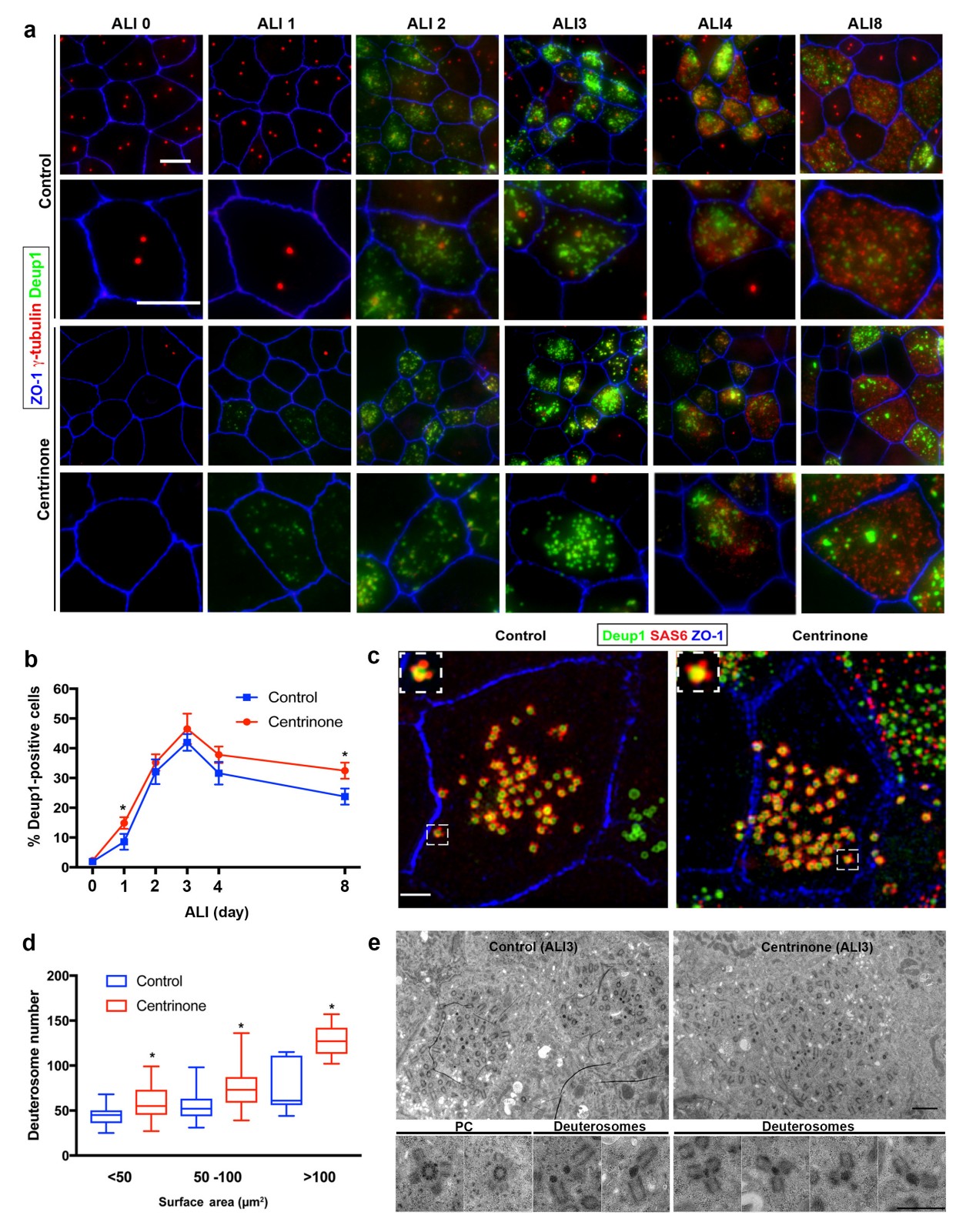

**Figure 3.** Deuterosome biogenesis is unaffected by loss of parental centrioles. (**a**) Immunofluorescence images of control and Centrinone-treated cells stained with antibodies against deuterosomes (Deup1), centrioles (γ-tubulin) and apical cell-cell junctions (ZO-1). Upper panels show lower magnification images, and lower panels highlight a single cell. Loss of PC does not disrupt deuterosome formation. In contrast, they appear earlier than in control cells (e.g. ALI 1) and persist for a longer period of time. Scale bar = 10 μm. (**b**) Quantification of the fraction of Deup1-expressing cells during

*Figure 3 continued on next page*

*Figure 3 continued*

differentiation. Results are averages of two independent experiments. More than 1000 cells were counted per sample for each time point. *p<0.05. (c) 3D-SIM images of MTEC at ALI stained for deuterosomes (Deup1), procentrioles (Sas6) and apical cell-cell junctions (ZO-1). Scale bar = 10 µm. (d) Quantification of deuterosome number in control and Centrinone-treated cells. Cells were grouped based on the size of their surface area, and the average number of deuterosomes per cell determined. Loss of PC causes an increase in deuterosome number in each category. N = 87 (control) and 71 (Centrinone). *p<0.05. (e) TEM images of control and Centrinone-treated MTEC at ALI3, confirming the presence of procentriole-forming deuterosomes that appear normal in size and morphology. Scale bars = 2 µm (upper panels) and 800 nm (magnified lower panels).

DOI: https://doi.org/10.7554/eLife.44039.008

The following figure supplement is available for figure 3:

**Figure supplement 1.** – Deuterosome-mediated procentriole formation is unaffected by loss of parental centrioles.

DOI: https://doi.org/10.7554/eLife.44039.009

control cells at ALI12 (*Figure 5i–j*). In sum, manipulating Plk4 protein levels does not reduce deuterosome or basal body number, nor does increasing the complement of parental centrioles.

## Centriole number scales with surface area

Since establishment of centriole abundance appears to be independent of the PC-deuterosome mediated pathway, we wondered if other cellular properties influenced centriole-cilia abundance. We discovered that airway multiciliated cells grown in vitro display a broad range in centriole number, from 100 to 600 centrioles per cell (*Figure 4*). We wanted to determine whether this broad range in centriole number, and its relationship to surface area, was retained in vivo. Trachea isolated from wild-type mice was cut longitudinally to expose the lumen, immunostained with centrin and ZO-1, then imaged by 3D-SIM (*Figure 6a*). Quantification of centriole number showed a similar variation in centriole abundance (*Figure 6b*). Consistent with the measurements in vitro, there was a direct relationship between centriole number and surface area, as larger MCC contained more centrioles compared to smaller cells (*Figure 6b–d*).

Next, we wondered if cell size (surface area or volume) was a major determinant of centriole abundance. In MTEC cultures, cell size and surface area are established roughly 1–2 days before the ciliogenesis transcriptional program is initiated. This temporal separation provided us with a means to determine whether the centriole amplification machinery responds to the differences in cell size set at the end of the proliferation phase. To test this, we manipulated cell shape and size prior to inducing differentiation by growing basal cells on varying concentrations of extracellular collagen. Modulating extracellular matrix density is a commonly used approach to manipulate cell volume and/or surface area (*Califano and Reinhart-King, 2010*; *Plant et al., 2009*; *Trappmann and Chen, 2013*; *Wells, 2008*). Culture and differentiation of basal cells on increasing concentrations of extracellular collagen resulted in a dose-dependent increase in cell surface area (*Figure 7a–b*). Remarkably, centriole abundance also increased in a dose-dependent manner (*Figure 7c*). To determine whether the increase in average centriole number per cell was a function of increased cell volume, we measured cell depth (apical-basal distance). Although the cells contained a larger surface area, they were shallower in depth, resulting in a similar overall cell volume (*Figure 7a and d–e*). Thus, the size of surface area, but not overall cell volume, appears to dictate centriole abundance during amplification. Finally, we sought to determine whether the higher centriole number in cells with larger surface areas was regulated at the level of the deuterosome. Indeed, cells grown on higher concentrations of collagen formed a higher proportion of deuterosomes per cell on average (*Figure 7f–g*). Collectively, these results indicate that a cell-intrinsic surface area-dependent process establishes centriole abundance, potentially by modulating deuterosome number.

## Discussion

In this study, we sought to determine how MCC establish the number of centrioles and cilia that each cell contains. We first tested the relative contributions of the PC and Plk4 to deuterosome formation, centriole amplification, and establishment of centriole number. Loss of PC had no impact on deuterosome formation and function indicating that, although deuterosomes can be nucleated from the original PC (*Al Jord et al., 2014*), their presence is not essential for the de novo centriologenesis pathway. In contrast, the average number of procentriole-forming deuterosomes was slightly but

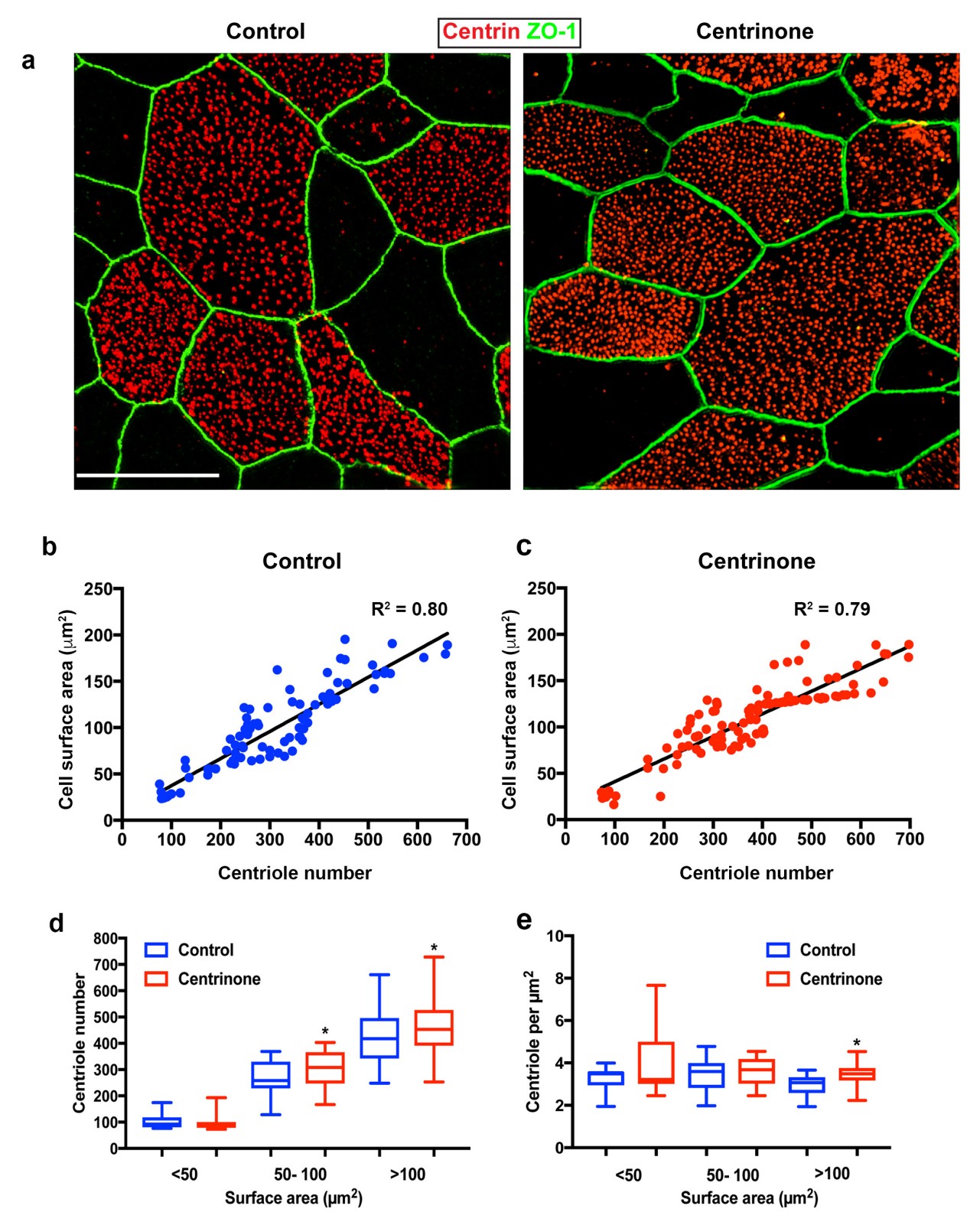

**Figure 4.** Centriole abundance correlates with surface area and is unaffected by loss of PC. (a) Example 3D-SIM images of mature multiciliated cells at ALI12 stained with markers of centrioles (centrin) and apical cell junctions (ZO-1). Scale bar = 10 μm. (b–c) Quantification of centriole number in control cells shows a linear relationship with cell surface area, such that larger cells contain more centrioles. Loss of the PC following Centrinone-treatment does not affect this relationship. (d–e) Comparison of abundance and density in mature MCC of varying sizes at ALI12. Loss of the PC did not result in a
*Figure 4 continued on next page*

*Figure 4 continued*

decrease in average centriole number (**d**). In fact, there was a slight increase in some cells. Similarly, the density and distribution of these centrioles at the apical surface was generally unchanged (**e**), although we noted a small but significant increase in the density of centrioles, possibly due to the elevated number per cell. Results are averages of three independent experiments. N = 91 (control) and 114 (Centrinone). *p<0.05.
DOI: https://doi.org/10.7554/eLife.44039.010

significantly higher upon loss of the PC (*Figures 3* and *5*). One intriguing possibility for the increase may be that the deuterosome pathway compensates for the loss of PC-derived centrioles by increasing deuterosome-mediated centriole assembly. This is consistent with our observation that cells lacking the PC formed a slightly higher number of centrioles on average when fully mature (*Figures 4–5*). Similarly, inhibition of deuterosome formation was shown to result in increased procentriole nucleation by the PC (*Zhao et al., 2013*), suggesting that these two centriole assembly pathways likely communicate to regulate centriole biogenesis. However, loss of the PC did impact the kinetics of deuterosome formation; they were evident in PC-less cells slightly earlier than control cells, and persisted at stages (e.g. ALI8) when they would normally have disappeared (*Figures 3* and *5*). We interpret these data to suggest that, although the PC are not critical for deuterosome biogenesis, they may help to control the number of deuterosomes formed, as well as the timing of their assembly and disassembly. During the revision of the manuscript, two studies (one preprint [*Mercey et al., 2018*] and one published [*Zhao et al., 2019*]) report similar parental centriole-independent deuterosome formation in Centrinone-treated ependymal multiciliated cells. Altogether, these data indicate that deuterosome formation and initiation of centriole amplification can proceed normally is the absence of PC.

Loss of parental centrioles (and thus, centrosomes) in dividing retinal pigment epithelial cells has been shown to trigger a p53-mediated stress response that inhibits their proliferation (*Fong et al., 2016*; *Lambrus et al., 2016*; *Meitinger et al., 2016*). Yet, loss of the PC in airway basal cells did not block or slow their growth, suggesting that the downstream consequences may differ in various cells or tissues. Phenotypes of patients with mutations in centrosomal genes support this notion. For example, mutations in components of the centriole assembly pathway are a major cause of microcephaly (*Jayaraman et al., 2018*). These mutations disrupt the proliferation and differentiation program of neuronal precursors, leading to p53-mediated loss of the progenitor pool, resulting in the small brain phenotype (*Nano and Basto, 2017*). In contrast, mutations in centrosome duplication factors can also cause cystic and fibrotic kidney diseases (*Reiter and Leroux, 2017*; *Srivastava et al., 2017*), which are characterized in part by enhanced and sustained cell proliferation. Therefore, the response of cells to centrosome loss, and the p53-dependent mitotic surveillance mechanism that senses this defect, may be cell- and tissue specific.

Although Plk4 kinase function was needed for canonical centriole duplication in proliferating basal cells, it was dispensable for post-mitotic centriole amplification. This was somewhat surprising for two reasons: (a) expression levels of Plk4 increase significantly during centriole amplification in MCC (*Hoh et al., 2012*; *Zhao et al., 2013*), suggestive of a role during those stages, and (b) the kinase is essential for initiating procentriole formation in the majority of mammalian cells examined (*Nigg and Holland, 2018*; *Sillibourne and Bornens, 2010*). However, Plk4 is absent in a number of organisms that contain centrioles. For example, the unicellular ciliates *Chlamydomonas* and *Tetrahymena* are able to assembly and duplicate their centrioles (basal bodies) without Plk4 (*Carvalho-Santos et al., 2011*; *Dutcher and O'Toole, 2016*). Similarly, Plk4-interacting proteins such as STIL and Cep152 are missing in a number of organisms with centrioles (*Carvalho-Santos et al., 2011*), highlighting the presence of other mechanisms that allow for control of centriole formation. Thus, even though components of the centriole assembly machinery are generally conserved throughout evolution, our results suggest that certain mammalian cell types may have adapted mechanisms to initiate centriologenesis independent of Plk4 kinase function.

Depletion of Plk4 in MTEC did cause a delay in centriologenesis, indicating that the protein itself might be needed for proper progression through the various stages of centriole assembly. This is reminiscent of what was recently described for other kinases involved in coordinating centriole assembly and cell cycle progression in MCC. For example, it was shown that differentiating, non-dividing MCC repurpose the mitotic regulatory circuitry involving CDK1/Plk1/APC-C to control the

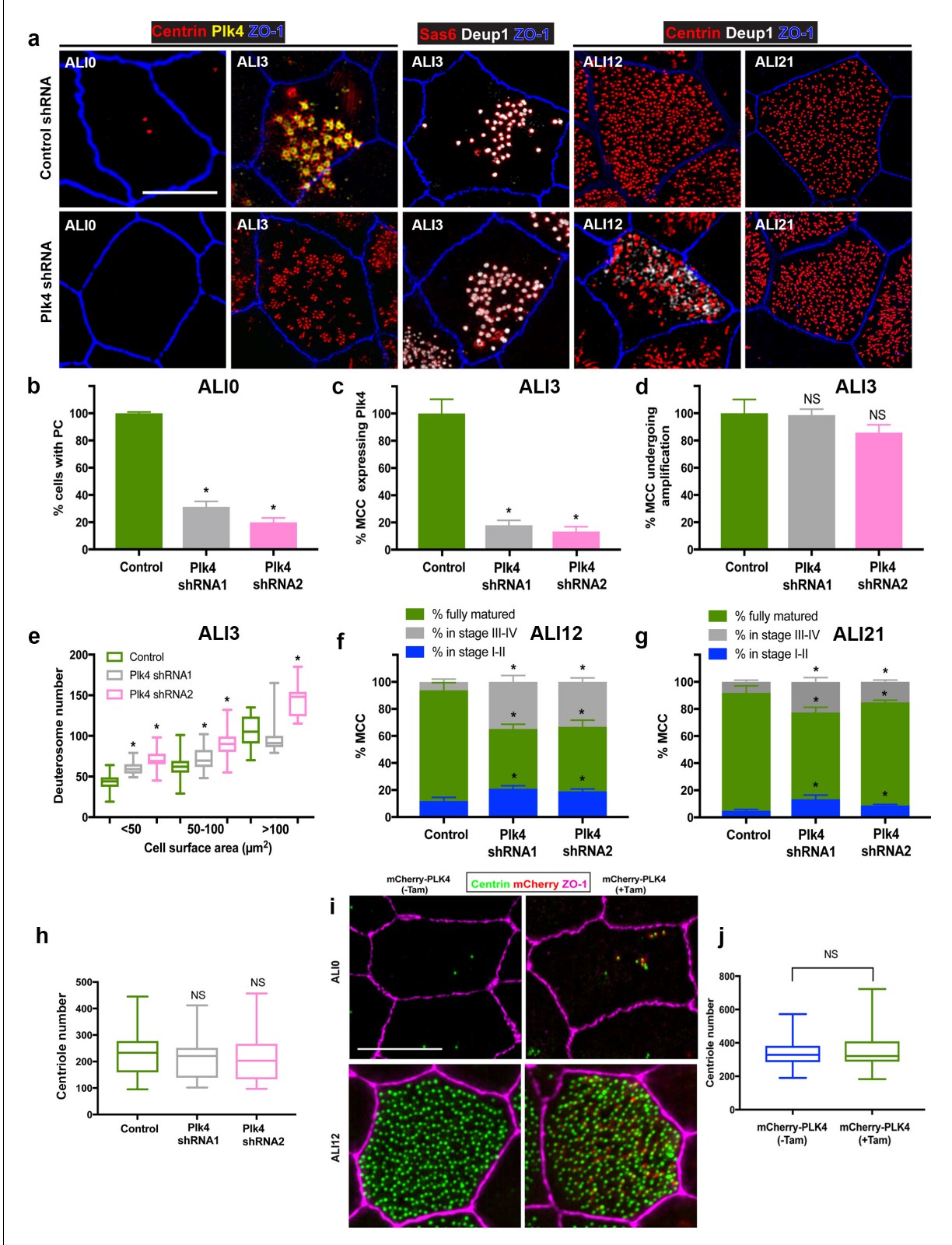

**Figure 5.** Depletion of Plk4 causes a delay in centriologenesis but does not affect abundance. (a) Images of MTEC infected with control or Plk4-targeting shRNA. Cells were fixed on the indicated days and stained for centrioles (centrin), endogenous Plk4, procentrioles (Sas6 or centrin), deuterosomes (Deup1) and apical cell junctions (ZO-1). Scale bar = 10 μm. (b) Depletion of Plk4 in basal cells causes loss of parental centrioles by ALI0, quantified using centrin staining of PC (a). Results are averages of three independent experiments. N = 862 (control shRNA), 846 (Plk4 shRNA1) and

*Figure 5 continued on next page*

*Figure 5 continued*

1108 (Plk4 shRNA2). *p<0.05. (c–d) Fraction of MCC at ALI3 undergoing centriole amplification in the absence of Plk4. The percentage of cells that initiated centriole amplification, determined using Deup1 and centrin staining, was unchanged (d). However, there was a large decrease in the fraction of MCC with detectable Plk4 (c). Results are averages of two independent experiments. N = 4158 (control shRNA), 1596 (Plk4 shRNA1) and 3456 (Plk4 shRNA2). *p<0.05. (e) Quantification of deuterosome number in control and Plk4-depleted cells. Similar to Centrinone-mediated PC loss, the number of deuterosomes per cell increases in the absence of PC. Results are averages of two independent experiments. N = 43 (control shRNA), 59 (Plk4 shRNA1) and 67 (Plk4 shRNA2) cells. *p<0.05. (f–g) Comparison of centriole amplification stages upon Plk4 depletion at ALI12 (f) and ALI21 (g). Although the proportion of the MCC in the population was unchanged in cells lacking Plk4, a significant fraction of the cells were in Stage I-II (with Deup1-positive deuterosomes still evident) and Stage III-IV (absence of deuterosomes) of centriole amplification (f). Culturing the cells for an additional 9 days (ALI21) resulted in an increase in the percentage of mature MCC with cilia in Plk4-depleted cells (g), highlighting a delay in the maturation process. Results are averages of two independent experiments. N = 815 (control shRNA), 783 (Plk4 shRNA1) and 1346 (Plk4 shRNA2). *p<0.05. (h) Comparison of centriole abundance in control and Plk4-depleted mature MCC at ALI21. Loss of Plk4 and PC did not result in a decrease in average centriole number. N = 43 (control shRNA), 57 (Plk4 shRNA1) and 67 (Plk4 shRNA2). *p<0.05. (i) 3D-SIM images of MTEC generated from Tg::mChPlk4/Rosa26-Cre[ERT2] (described in detail in Materials and methods), stained for exogenous Plk4 (mCherry), centrioles (centrin) and apical cell junctions (ZO-1). Addition of tamoxifen during proliferation resulted in constitutive overexpression of mChPlk4,and caused formation of supernumerary PC by ALI0. Importantly, the protein was continuously expressed and still evident in fully mature MCC at ALI12. Scale bar = 10 μm. (j) Supernumerary parental centrioles and constitutive overexpression of Plk4 do not result in increased centriole abundance in mature MCC. Results are averages of two independent experiments. N = 108(-TAM), and 105 (+TAM).

DOI: https://doi.org/10.7554/eLife.44039.011

The following figure supplement is available for figure 5:

**Figure supplement 1.** – Combined depletion and Centrinone-mediated inhibition of Plk4 causes a delay in centriologenesis.

DOI: https://doi.org/10.7554/eLife.44039.012

timely progression of centriole amplification, maturation, and motile ciliogenesis while avoiding reentry into mitosis (*Al Jord et al., 2017*). Another study found that CDK2, the kinase responsible for G1-S phase transition, was also required in MCC to initiate the motile ciliogenesis program independent of cell cycle progression (*Vladar et al., 2018*). Thus, one possible reason for the elevated Plk4 protein is to coordinate the timing of centriole assembly and maturation in post-mitotic cells. Consistent with this theory, a recent study in *Drosophila* identified a role for Plk4 in regulating the rate and period of procentriole growth (*Aydogan et al., 2018*), demonstrating that Plk4 functions as a homeostatic clock to ensure centrioles grow to the correct size. Indeed, we found that MCC lacking Plk4 initiated centriole assembly to the same extent as control cells, were delayed in passage through the growth and maturation phases, but eventually 'caught up' (*Figure 5*). Importantly, multiciliated cells lacking Plk4 contained the same number of centrioles on average when fully mature at ALI21, further indicating that it is not critical for regulating number per se. Moreover, overexpression of Plk4 in MTEC (*Figure 5*) or in *Xenopus* larvae MCC (*Klos Dehring et al., 2013*) did not result in increased centriole number. Thus, Plk4 may play a similar role as CDK1/CDK2/Plk1/APC-C, by participating in a temporal regulatory mechanism that mediates passage through the various centriole assembly steps.

Centriole abundance in MCC scales with surface area, a phenomenon we observed in airway tissues in vivo and in MTEC cultures in vitro. However, it is unclear which of those properties influences the other: does having a larger surface area result in the formation of more centrioles, or does a cell that forms a larger number of centrioles expand its surface area to accommodate them? One advantage of using the MTEC culture system is that the ciliogenesis program initiates roughly 2 days after basal cells have already established their size and surface area at ALI0. Therefore, we could temporally separate these two events. By growing cells on increasing extracellular collagen matrix density during the proliferation phase, we caused the enlargement of cell surface area before the transcriptional ciliogenesis program initiated. We discovered that cells formed more centrioles once fully differentiated, suggesting that the centriole amplification machinery responds to the change in surface area. We attempted the reciprocal experiment, which was to induce the formation of excess centrioles and test whether the size of the surface area changed accordingly. Although constitutive overexpression of Plk4 did result in the formation of excess PC, it did not alter final centriole number or surface area (*Figure 5* and data not shown).

How are the variations in cell surface area communicated to the centriole amplification pathway to establish centriole number? There are at least three possible ways we envision this could occur. First, larger cells might increase transcription of genes essential for centriole and cilia assembly. This

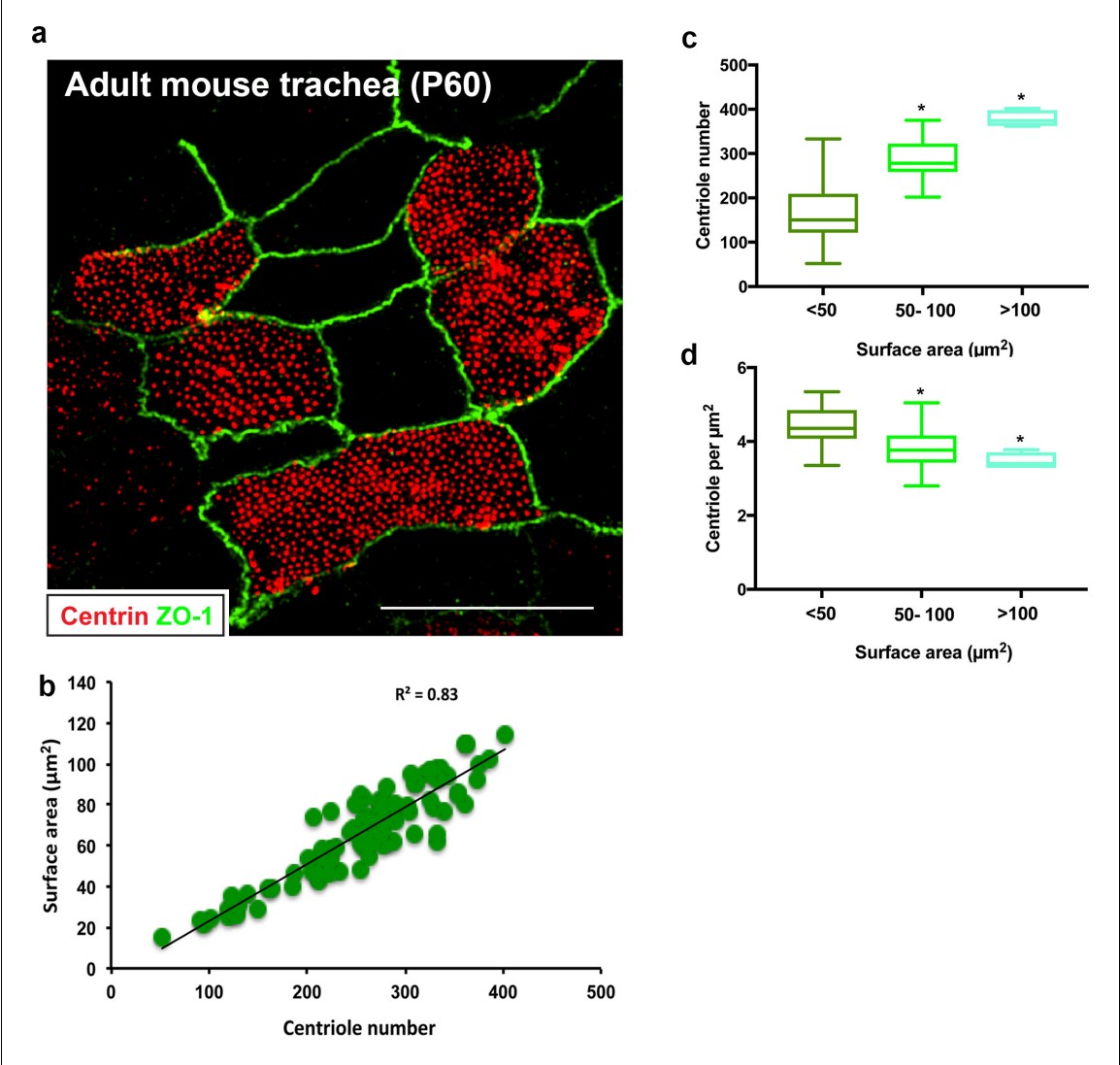

**Figure 6.** Centriole abundance correlates with surface area in vivo. (**a**) Example 3D-SIM image of adult mouse trachea (**P60**) immunostained for centrioles (centrin) and apical cell junctions (ZO-1). Scale bar = 10 μm. (**b–d**) Centriole number ranges between 100 and 400 per cell on average and displays a liner relationship with surface area, consistent with the observations in MTEC. Similarly, centriole density decreases slightly as surface area increases. N = 95 cells.

DOI: https://doi.org/10.7554/eLife.44039.013

would be analogous to the 'limiting component' model of organelle abundance (*Chan and Marshall, 2012*; *Goehring and Hyman, 2012*; *Marshall, 2016*), where a fixed quantity of a precursor protein (s) would be expressed then 'used up' as centriole assembly occurs. In this scenario, the number of centrioles assembled would stop once the limiting component is no longer available. We did note an increase in deuterosome number in cells with enlarged surface area (*Figure 7*), suggesting that transcriptional output of at least some precursors is likely elevated. However, it remains unclear whether there is a single limiting component, or if all centriolar proteins become expressed at higher levels in larger cells. Alternatively, cells with larger surface areas might spend a longer period of time in stages of centriole assembly compared to smaller cells. In this scenario, the rate of centriole assembly would be the same in cells of different size, but ones with larger surface areas would spend longer periods in stages I-III to generate more total centrioles. Since the amount of time spent in these assembly stages is coordinated by proteins such as CDK1/Plk1/CDK2/APC-C (*Al Jord et al., 2017*; *Vladar et al., 2018*), one possibility is that these pathways help relay information about cell

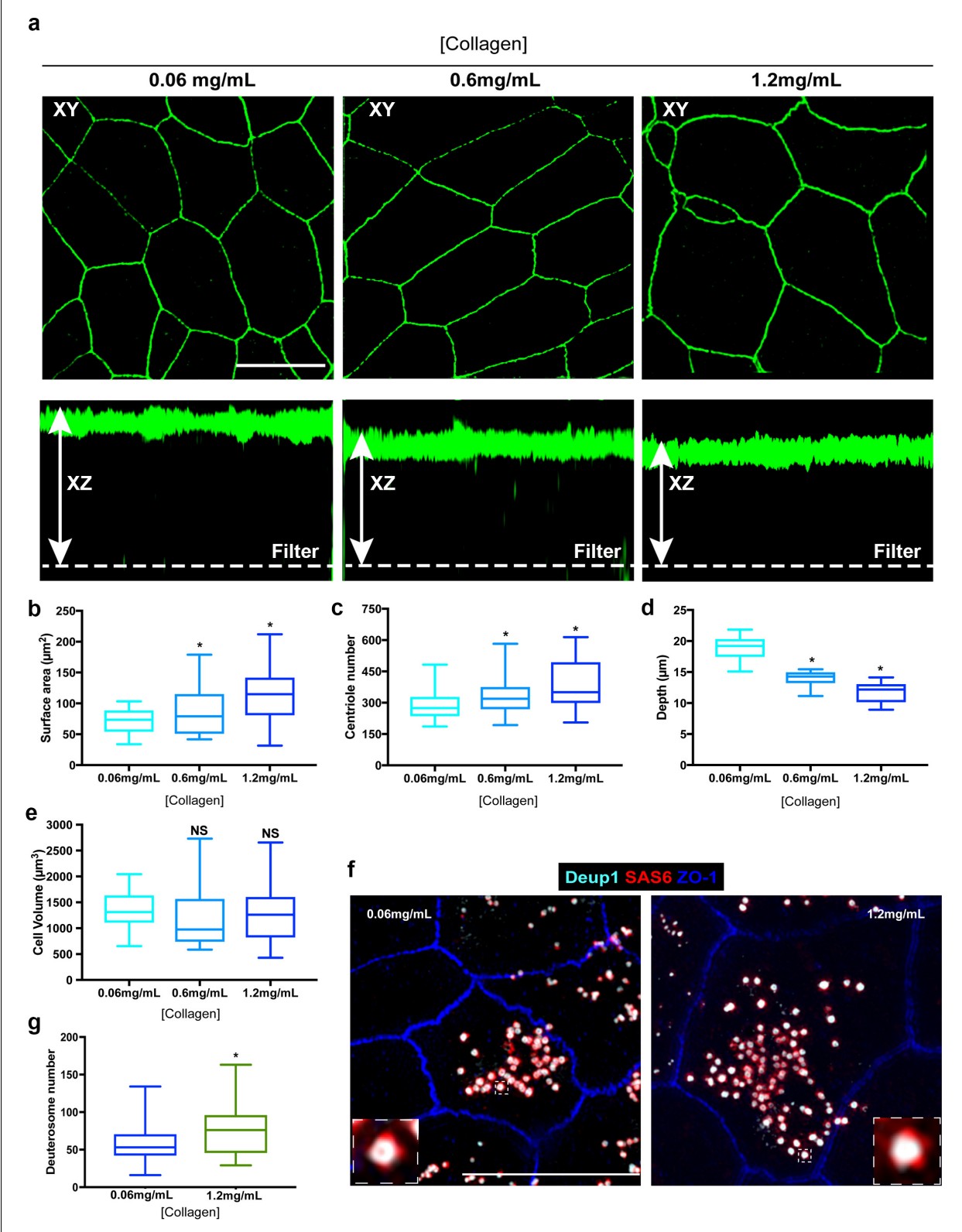

**Figure 7.** Enlargement of the surface area of progenitor cells results in increased centriole abundance. (**a**) Example 3D-SIM images of MTEC grown on varying concentrations of extracellular collagen matrix. Top panels show X-Y orientation, and lower panels the X-Z dimension. Scale bar = 10 μm. (**b–e**) Analysis of surface area (**b**), centriole number (**c**), apical-basal distance (depth, (**d**) and cell volume (**e**) in MTEC cultured on differing concentrations of collagen. Increasing the collagen density resulted in a dose-dependent expansion of the apical surface, whereas the cells became shallower, resulting

*Figure 7 continued on next page*

*Figure 7 continued*

in a similar overall cell volume. Importantly, the increase in surface are at ALI0 resulted in increased centriole number (**c**). Results are averages of two independent experiments. N = 39 (0.06mg/mL), 48 (0.6 mg/mL) and 42 (1.2mg/mL). *p<0.05. (**f**) 3D-SIM images of MTEC grown on normal (0.06 mg/mL) and high (1.2 mg/mL) levels of extracellular collagen. Cells at ALI3 were immunostained with antibodies to mark deuterosomes (Deup1), procentrioles (Sas6) and apical cell junctions (ZO-1). Scale bar = 10 µm. (**g**) Quantification of deuterosome number in MTEC at ALI3 showed a significant increase in cells grown on increased collagen density (with larger surface areas). Results are averages of two independent experiments. N = 45 (0.06mg/mL) and 43 (1.2mg/mL). *p<0.05.

DOI: https://doi.org/10.7554/eLife.44039.014

surface area/size to the centriole assembly machinery, and fine-tune the length of time spent in assembly to achieve the desired final number.

Finally, it is possible that the apical cytoskeleton helps instruct the centriole amplification pathway to regulate centriole abundance per cell. It is well established that an apical actin network in MCC plays a critical role in modulating the surface area, regulating centriole migration and docking at the plasma membrane, ensuring the even distribution of centrioles across the cell surface, and orienting (planar polarization) of centrioles (*Antoniades et al., 2014*; *Herawati et al., 2016*; *Kulkarni et al., 2018*; *Mahuzier et al., 2018*; *Pan et al., 2007*; *Sedzinski et al., 2016*; *Sedzinski et al., 2017*; *Werner et al., 2011*). What remains unclear is whether this actin network communicates with the centriole amplification machinery to regulate centriole number. One possibility is that the variations in surface area may result in corresponding changes to the size of the apical actin lattice, which could then instruct the centriologenesis program to generate more centrioles. Our future studies will focus on defining the pathway(s) coordinating the relationship between cell surface area and centriole amplification. These will have important clinical implications for ciliopathies caused by reduced cilia abundance in MCC, such Primary Ciliary Dyskinesia and hydrocephalus.

## Materials and methods

### Cell culture

All animal studies were performed following protocols that are compliant with guidelines of the Institutional Animal Care and Use Committee at Washington University and the National Institutes of Health. Mouse Tracheal Epithelial Cell (MTEC) cultures were established as previously described (*Mahjoub et al., 2010*; *Silva et al., 2016*; *You and Brody, 2013*; *You et al., 2002*). Briefly, C57BL/6J mice were euthanized at 2–4 months of age, trachea were excised, opened longitudinally to expose the lumen, placed in 1.5 mg/ml pronase E in DMEM/F-12 supplemented with antibiotics and incubated at 4°C overnight. Tracheal epithelial progenitor cells were dislodged by gentle agitation and collected in DMEM/F12 containing 10% FBS. After centrifugation at 4°C for 10 min at 800 g, cells were resuspended in DMEM/F12 with 10% FBS and plated in a Primaria Tissue Culture dish (Corning) for 3–4 hr at 37°C with 5% $CO_2$ to adhere contaminating fibroblasts. Non-adhered cells were collected, concentrated by centrifugation, resuspended in an appropriate volume of MTEC-complete medium (described in *You and Brody, 2013*; *You et al., 2002*), and seeded at a density of $9 \times 10^4$ cells/cm$^2$ onto Transwell-Clear permeable filter supports (Corning) treated with 0.06 mg/ml rat tail Collagen type I. Air liquid interface (ALI) was established after cells reached confluence by feeding cells with MTEC serum-free medium (*You and Brody, 2013*; *You et al., 2002*) only in the lower chamber. To ablate parental centrioles using Centrinone, cells were incubated in medium supplemented with 1 µM Centrinone A (*Wong et al., 2015*) beginning at 2 days after seeding. Cells were cultured at 37°C with 5% $CO_2$, and media containing Centrinone replaced every 2 days for up to 21 days. Samples were harvested at ALI days 0, 1, 2, 3, 4, 6, 8, 12 and 21, fixed and analyzed as described below. All chemicals were obtained from Sigma-Aldrich unless otherwise indicated. All media were supplemented with 100 U/ml penicillin, 100 mg/ml streptomycin, and 0.25 mg/ml Fungizone (all from Invitrogen). Samples were tested for mycoplasma contamination using commercially available kits.

To induce constitutive overexpression of Plk4 in MTEC, transgenic Tg::mCherry-Plk4 animals containing a Chloramphenicol Acetyl Transferase (CAT) coding sequence that includes a stop codon flanked by two loxP sites (*Dionne et al., 2018*; *Marthiens et al., 2012*) were mated to Tg::Rosa26-

Cre[ERT2] mice (*Ventura et al., 2007*) that conditionally express Cre-recombinase in the majority of tissues. Trachea were isolated from Tg::mChPlk4-Rosa26-Cre[ERT2] and basal progenitor cells isolated as described. Cre-recombinase expression was induced by adding 1 μm 4-hydroxytamoxifen (Sigma) during the proliferation stage, beginning at 2 days post-seeding. Cells were incubated with tamoxifen for a total of 4–6 days up to ALI 0 to induce expression of mChPlk4. Air liquid-interface was established once cells reached full confluence, samples harvested and fixed at ALI days 0, 1, 2, 3, 4, 8, 12 as described above.

## Lentivirus production and cell infection

Two lentivirus-based shRNA constructs targeting mouse Plk4 (shRNA#1: nucleotides 726–747, 5'-GCTTTGACAATCTACCAGAAA −3'; shRNA#2: nucleotides 1896–1917, 5'- CCACCAGTTACTTCG TAGAAA −3') were used. Plasmids were obtained from the RNAi Consortium (TRC) collection available at the Washington University School of Medicine Hope Center for Neurological Disorders Viral Vectors Core (https://hopecenter.wustl.edu). Lentivirus was produced by co-transfection of HEK293T cells with the appropriate transfer and lentiviral helper plasmids (pCMVDR8.74 and pMD2.VSVG) using Lipofectamine 3000. The HEK293T cells were originally obtained from the ATCC and their identity authenticated by STR Profiling analysis. We have tested for mycoplasma contamination using commercially available kits. To infect MTEC, cells were seeded on Transwell filters in the presence of shRNA-expressing lentivirus at an MOI of 1, and infected cells selected with 2 μg/mL puromycin for 4–6 days until they reached confluence. ALI was established as described, and samples harvested on the indicated days. For experiments using a combination of shRNA-mediated depletion and Centrinone-mediated inhibition of Plk4, basal cells were infected as described above, cultured in the presence of 1 μM Centrinone, and samples harvested on the indicated days.

## Immunofluorescence

For immunofluorescence assays MTEC were rinsed twice with PBS and fixed in either 100% ice-cold methanol at −20℃ for 10 min, or with 4% paraformaldehyde in PBS at room temperature for 10 min, depending on antigen. Cells were rinsed twice with PBS, filters excised from their plastic supports and cut into quarters to provide multiple equivalent samples for parallel staining. Cells were briefly extracted with 0.2% Triton X-100 in PBS and blocked for 1 hr at room temperature with 3% BSA (Sigma) in PBS. Samples were incubated with primary antibodies for 1 hr at room temperature or 4℃ overnight. Primary antibodies used in the study: Mouse anti-Centrin (clone 20H5; 1:2,000; EMD Millipore 04–1624), rat anti-Cep120 (clone 2; 1:2000; *Betleja et al., 2018*), mouse anti-acetylated α-tubulin (clone 6-11B0-1; 1:5000; Sigma-T6793), rabbit anti-Cep164 (1:1000; *Firat-Karalar et al., 2014*), rabbit anti-Cep135 and rabbit anti-Cep152 (1:1000; gift from R Basto, Institute Curie, Paris, France), rabbit anti-Deup1 (1:400) and rabbit anti-Plk4 (1:200; *Zhao et al., 2013*), mouse anti-Sas6 (1:200; Santa Cruz - sc-81431), rabbit anti-STIL antibody (1:500; abcam-ab89314), rat anti-ZO-1 (1:1000; Santa Cruz sc-33725), rabbit anti-ZO-1 (1:500, Zymed Laboratories Inc - 61–7300). Alexa Fluor dye–conjugated secondary antibodies (Invitrogen) were used at a dilution of 1:500 at room temperature for 1 hr. Samples were mounted in Mowiol antifade medium containing N-propyl gallate (Sigma). For whole-mount in situ staining, freshly-excised trachea were fixed using 4% paraformaldehyde in PBS at room temperature for 5 min, then ice-cold methanol at −20℃ for 5 min. The tissue was washed in PBS, blocked and incubated with primary antibody.

Images were captured using a Nikon Eclipse Ti-E inverted confocal microscope equipped with a 60X (1.4 NA) or 100X (1.45 NA) Plan Fluor oil immersion objective lens (Nikon, Melville, NY). A series of digital optical sections (z-stacks) were captured at 0.2 μm intervals using an Andor Neo-Zyla CMOS camera at room temperature. Three-dimensional superresolution structured illumination microscopy (3D-SIM) images were captured on an inverted Nikon Ti-E microscope using 100X oil objective (1.45 NA) at the Washington University Center for Cellular Imaging. Optical z-sections were separated by 0.216 μm, and 3D-SIM images were processed and reconstructed using the Nikon Elements AR 4 Software.

## Electron microscopy

MTEC grown on Transwell filters were fixed with freshly prepared 2.5% glutaraldehyde in PBS, washed in PBS for 30 min, pre-stained with 2% osmium tetroxide and 1% uranyl acetate, dehydrated

in graded ethanol series, and then embedded in Embed 812 resin. Samples were serially sectioned from the cell's apical surface to the basal membrane, above the filter. The thickness of the sections was 80 nm (for ALI3 samples), or 120 (for ALI0 samples). Serial sections were transferred onto formvar coated copper slot grids, stained with 2% uranyl acetate and 0.4% lead citrate, and imaged using a transmission electron microscope (FEI Tecnai T12) operating at 80 kV. The alignment of the serial sections and image analysis was performed in Adobe Photoshop and Fiji (NIH).

### Image analysis and measurements

Quantification of centriole abundance, deuterosome number, surface area and volume were performed using Nikon Elements AR 4 Software. To determine the extent of parental centriole loss, cells at ALI 0 were scored for centriole number (0, 1, 2, 3, four centrioles) per cell. To determine the fraction of cells in the population undergoing centriole amplification during differentiation, cells at ALI 1, 2, 3, 4 and 8 were scored for the presence of centriolar aggregates (Agg.), which then become defined as multiciliated cells (MCC). This approach was applied to each centriolar marker, and the percentage of the population undergoing centriole amplification determined. The surface area was defined by staining cells with ZO-1 to mark the apical cell-cell junctions, and measured using the thresholding algorithm (Elements AR 4). Cells were binned into three categories that ranged from what we define as small (<50 $\mu m^2$), medium (50–100 $\mu m^2$) and large (>100 $\mu m^2$). Cell volume was determined by measuring the apical-basal distance (depth), which was then multiplied by the surface area. To quantify deuterosome number, MCC at ALI 3–4 were stained for markers of the deuterosome and procentrioles. The number of procentriole-forming deuterosome rings/rosettes were scored and compared in cells of similar size. The number of procentrioles per deuterosome was calculated from 3D-SIM images. To determine the number of centrioles per cell in mature MCC at ALI12 or 21, the spot detection tool (Elements AR 4) with a spot circular ROI of 0.26 $\mu m$ was used. Centriole abundance was then grouped based on cell size as described. Centrioles density was determined by dividing the number of centrioles per cell by the surface area.

### Manipulation of cell surface area

Freshly isolated basal cells were seeded onto Transwell filters coated with varying concentrations (from 0.06 mg/mL – 2.4 mg/mL) of type one rat tail collagen. Cells were cultured for 4–6 days until confluent then ALI established. Samples were fixed on the indicated days as analyzed for centriole number, surface area and volume. We found that the size of the surface area did not increase significantly beyond 1.2 mg/mL collagen concentration, thus we set that value as the upper limit for all subsequent analyses.

### Statistical analyses

Statistical analyses were performed using GraphPad PRISM 7.0 or Microsoft Excel. The vertical segments in box plots show the first quartile, median, and third quartile. The whiskers on both ends represent the maximum and minimum values for each dataset analyzed. For bar graphs, data are reported as mean ± SEM. Collected data were examined by two-tail unpaired Student's t-test. Statistical significance was set at $p < 0.05$.

### Acknowledgements

We thank X Zhu (Shanghai Institutes for Biological Sciences, China) for generously sharing the Deup1 and Plk4 antibodies, N Da Silva and R Basto (CNRs, Institut Curie, Paris, France) for providing the Tg::mChPlk4 mice, Cep152 and Cep135 antibodies, K Oogema and AK Shiau (Ludwig Institute for Cancer Research, La Jolla, CA) for initially sharing the Centrinone compounds. We thank members of the Washington University Center for Cellular Imaging (WUCCI) for assistance with some image acquisition and data analyses. We also thank S Dutcher, P Bayly, A Horani and all members of the Washington University Cilia Group for critical reading of the manuscript. This study was supported by funding from the National Institute of Diabetes and Digestive and Kidney Diseases (R01-DK108005) to MRM, and the National Heart, Lung and Blood Institute (R01-HL128370) to MRM and SLB. The authors declare no competing financial interests.

## Additional information

### Funding

| Funder | Grant reference number | Author |
|---|---|---|
| National Heart, Lung, and Blood Institute | R01-HL128370 | Steven L Brody<br>Moe R Mahjoub |
| National Institute of Diabetes and Digestive and Kidney Diseases | R01-DK108005 | Moe R Mahjoub |

The funders had no role in study design, data collection and interpretation, or the decision to submit the work for publication.

### Author contributions

Rashmi Nanjundappa, Data curation, Formal analysis, Validation, Investigation, Visualization, Methodology, Writing—original draft, Writing—review and editing; Dong Kong, Formal analysis, Investigation, Methodology; Kyuhwan Shim, Investigation, Methodology, Writing—review and editing; Tim Stearns, Steven L Brody, Conceptualization, Resources, Writing—review and editing; Jadranka Loncarek, Formal analysis, Investigation, Methodology, Writing—review and editing; Moe R Mahjoub, Conceptualization, Supervision, Funding acquisition, Investigation, Writing—original draft, Project administration, Writing—review and editing

### Author ORCIDs

Rashmi Nanjundappa (iD) http://orcid.org/0000-0003-3621-4628
Tim Stearns (iD) http://orcid.org/0000-0002-0671-6582
Moe R Mahjoub (iD) http://orcid.org/0000-0001-8129-7464

### Ethics

Animal experimentation: This study was performed in strict accordance with the recommendations in the Guide for the Care and Use of Laboratory Animals of the National Institutes of Health. Moreover, the experiments were performed following approved protocols that are compliant with guidelines of the Institutional Animal Care and Use Committee at Washington University (approval # 20180237). Mice were euthanized using carbon dioxide inhalation followed by cervical dislocation, and every effort was made to minimize suffering and distress.

### Decision letter and Author response

Decision letter https://doi.org/10.7554/eLife.44039.017
Author response https://doi.org/10.7554/eLife.44039.018

## Additional files

### Supplementary files

• Transparent reporting form
DOI: https://doi.org/10.7554/eLife.44039.015

### Data availability

All data generated or analysed during this study are included in the manuscript.

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
