## [Decision Letter]

Thank you for submitting your article "Regulation of cilia abundance in multiciliated cells" for consideration by *eLife*. Your article has been reviewed by three peer reviewers, and the evaluation has been overseen by a Reviewing Editor and Anna Akhmanova as the Senior Editor. The following individual involved in review of your submission has agreed to reveal their identity: Meng-Fu Bryan Tsou (Reviewer #1).

The reviewers have discussed the reviews with one another and the Reviewing Editor has drafted this decision to help you prepare a revised submission.

Summary:

This work investigates mechanisms controlling basal body abundance in multiciliated cells (MCCs). Previous work had indicated that the daughter centriole and deuterosomes participate in the generation of MCC basal bodies. PLK4, a previously identified master regulator of centriole assembly, was functionally assessed using centrinone, a pharmacological PLK4 inhibitor, or Plk4 knockdown. Notably, PLK4 inhibition did inhibit duplication of the parental centrioles, but did not affect total basal body number in MCCs, suggesting that centrioles are not required for deuterosome formation or basal body/centriole/centrosome generation in MCCs. In the search for regulators of centriole number, perhaps independent of PLK4, the authors identified that centriole number scales with the size of the apical domain of cells. Scaling was already evident at the stage of deuterosome assembly, as cells with a larger surface formed more deuterosomes.

Essential revisions:

More evidence to support the Plk4 loss of function data is required to strengthen the paper. In particular, showing that known Plk4 substrates are normally recruited at centrioles following inactivation of Plk4 in MCCs will help support the author's view that the delay in centriole assembly following shRNA inactivation is due to other Plk4 functions (i.e. coupling to the cell cycle machinery).

In particular, it will be important to use a combination of the centrinone (1µM) and shRNA to fully inactivate PLK4. The authors have shown that shRNAi can reduce PLK4 to nearly a non-detectable level, so whatever the remaining level of PLK4 should be sufficiently inhibited by 1µM of the drug. The centriole biogenesis dynamics under this condition should be monitored and compared to the single treatment. Higher concentration of centrinone can also be tried, but 10 µM centrinone may potentially cause off target effects, leading to non-specific blocks of normal MCC development.

Also, loss of PLK4 should lead to loss of the key PLK4 substrate, STIL, at centrioles. The authors should check whether STIL is present in PLK4 loss of function centrioles. It will be striking to see STIL negative centrioles in MCCs, a strong support for the authors' conclusions.

Minor points:

1) Centrosome loss has been shown to induce p53-dependnet stress response in cycling cells, but it does not seem to occur in the MTEC system used here. If space is allowed, the author should perhaps discuss or highlight this interesting difference seen in basal cells of MCC.

2) The authors showed that centriole number is not affected in MCC lacking pre-existing centrioles. We wonder, in their EM studies, if the authors also check the structural integrity, or the shape, of these de-novo centrioles formed in the absence of parental centrioles and PLK4.

3) Plk4 over-expression in other cell types increases the number of procentriole assembly sites around parental centrioles, leading to the formation of "rosettes" (Habedanck et al., 2005). In MCCs, are rosettes already present, raising the possibility that all the possible assembly sites are already being used? Do the authors think that something similar could be happening for deuterosome-based centriole assembly (i.e. the formation of a given number of assembly sites independently of Plk4 and the use of all of them in normal conditions)? If so, Plk4 over-expression may not be sufficient to increase centriole number even if it does play a role in centriole assembly.

4) An interesting notion is that PLK4, STIL and the PLK4 scaffold CEP152/CEP192 are actually *not* conserved in all organisms that form centrioles. It would be interesting to have a paragraph discussing which organisms have centrosomes but not PLK4, STIL or CEP152/CEP192.

---

## [Author Response]

Essential revisions:More evidence to support the Plk4 loss of function data is required to strengthen the paper. In particular, showing that known Plk4 substrates are normally recruited at centrioles following inactivation of Plk4 in MCCs will help support the author's view that the delay in centriole assembly following shRNA inactivation is due to other Plk4 functions (i.e. coupling to the cell cycle machinery).In particular, it will be important to use a combination of the centrinone (1µM) and shRNA to fully inactivate PLK4. The authors have shown that shRNAi can reduce PLK4 to nearly a non-detectable level, so whatever the remaining level of PLK4 should be sufficiently inhibited by 1µM of the drug. The centriole biogenesis dynamics under this condition should be monitored and compared to the single treatment.

As recommended by the reviewers, we used a combination of shRNA-mediated depletion and Centrinone-mediated inhibition of Plk4 in our MTEC system. The dynamics of centriole biogenesis were examined as was described for the other experiments in the manuscript. Simultaneous depletion and Centrinone-mediated inhibition of Plk4 in basal cells caused loss of parental centrioles by ALI0, as expected. We found that the fraction of cells lacking Plk4 that initiated centriole assembly was unchanged, consistent with the centrinone and shRNA-mediated depletion experiments alone. Similar to the Plk4-depletion experiments, a significant fraction of the cells at ALI12 were immature, delayed in Stages I-II and III-IV of centriole amplification. Culturing the cells for an additional 9 days (ALI21) resulted in an increase in the percentage of mature cells, highlighting a delay in the steps of centriologenesis. Importantly, there was no additive effect: the addition of Centrinone to Plk4-shRNA infected cells did not block centriole amplification, but merely caused a delay in the process. This result is almost identical to the shRNA-based depletion results (Figure 5). Thus, we conclude that Plk4 is indeed dispensable for initiation of centriole amplification. These data are presented as new Figure 5—figure supplement 1, and added to the Results section (subsection “Modulating Plk4 expression delays centriole assembly but does not affect number”, first paragraph). We thank the reviewers for suggesting this experiment, which further adds to the rigor of our results.

Also, loss of PLK4 should lead to loss of the key PLK4 substrate, STIL, at centrioles. The authors should check whether STIL is present in PLK4 loss of function centrioles. It will be striking to see STIL negative centrioles in MCCs, a strong support for the authors' conclusions.

We examined the dynamics of STIL expression and localization in MTEC upon Plk4 inhibition (Centrinone) or depletion (shRNA plus Centrinone). Intriguingly, we discovered that STIL is expressed and recruited in a timely manner to sites of procentriole assembly during centriole amplification in Centrinone-treated cells (new Figure 2C). This suggests that parental centrioles and Plk4 kinase activity are dispensable for STIL localization. This is consistent with our analysis of another Plk4 substrate, Cep152, which we also showed is recruited normally (Figure 2E and Figure 2—figure supplement 1B). Moreover, STIL was localized at sites of procentriole assembly upon simultaneous depletion and inhibition of Plk4 (new Figure 5—figure supplement 1G-H). Therefore, it appears that the enrichment of centriole assembly factors to sites of procentriole formation (deuterosomes) is Plk4-independent during the centriole amplification stage of multiciliated cells.

We note that multiple organisms that assemble centrioles lack Plk4, while others appear to be missing its interactors such as STIL and Cep152. Thus, there are mechanisms that allow for control of centriole formation independent of the Plk4-STIL-Cep152 axis. Our results indicate that specialized mammalian cell types may have adapted mechanisms to initiate centriologenesis independent of Plk4 kinase function.

Minor points:1) Centrosome loss has been shown to induce p53-dependnet stress response in cycling cells, but it does not seem to occur in the MTEC system used here. If space is allowed, the author should perhaps discuss or highlight this interesting difference seen in basal cells of MCC.

We have highlighted this difference in a new paragraph in the Discussion section as follows:

“Loss of parental centrioles (and thus, centrosomes) in dividing retinal pigment epithelial cells has been shown to trigger a p53-mediated stress response that inhibits their proliferation (Fong et al., 2016; Lambrus et al., 2016; Meitinger et al., 2016). […] Therefore, the response of cells to centrosome loss, and the p53-dependent mitotic surveillance mechanism that senses this defect, may be cell- and tissue specific.”

2) The authors showed that centriole number is not affected in MCC lacking pre-existing centrioles. We wonder, in their EM studies, if the authors also check the structural integrity, or the shape, of these de-novo centrioles formed in the absence of parental centrioles and PLK4.

We attempted to address this question by examining the morphology and size (length, diameter etc) of procentrioles from our EM experiments. However, it is difficult to quantitatively compare centrioles between control and centrinone-treated samples by EM. This is due to the fact that the centrioles in each section are oriented differently, and we could not confidently select procentrioles that were in the exact same stage of growth, as well as orientation relative to the EM grid. Nonetheless, we note in the manuscript that the centrioles generated by centrinone-treated cells are functionally normal: cells form the correct number of centrioles (Figure 4), these centrioles mature into basal bodies (Figure 2—figure supplement 3A-B), dock at the plasma membrane and form cilia (Figure 3C-D), and these cilia beat normally (data not shown). Collectively, these results indicate that the integrity and functions of these de novo centrioles are unaffected by loss of parental centrioles.

3) Plk4 over-expression in other cell types increases the number of procentriole assembly sites around parental centrioles, leading to the formation of "rosettes" (Habedanck et al., 2005). In MCCs, are rosettes already present, raising the possibility that all the possible assembly sites are already being used? Do the authors think that something similar could be happening for deuterosome-based centriole assembly (i.e. the formation of a given number of assembly sites independently of Plk4 and the use of all of them in normal conditions)? If so, Plk4 over-expression may not be sufficient to increase centriole number even if it does play a role in centriole assembly.

In an effort to determine the status of available assembly sites on deuterosomes, we quantified the number of procentrioles formed by deuterosomes in control and PC-less (centrinone-treated) cells. We found that deuterosome “rosettes” in control cells formed an average of 5 procentrioles each at ALI3. The number of procentrioles per deuterosome ranged from 2-8 (new Figure 3—figure supplement 1).

Importantly, deuterosomes of centrinone-treated cells showed the exact same range, with an upper limit of 8 procentrioles per deuterosome. Thus, it is possible that a proportion of procentriole assembly sites are already being used, limiting the effects of Plk4 overexpression. However, most of the deuterosomes were forming less than the upper limit of 8 procentrioles (new Figure 4), and many deuterosomes in PC-less cells contained “open sites” for assembly. These results indicate that free sites for centriole assembly are likely available in the deuterosome population. Since overexpression of Plk4 did not cause an increase in centriole number in our MTEC (Figure 5I-J), nor in *Xenopus* multiciliated cells (Klos Dehring et al., 2013), we conclude that Plk4 is not rate limiting for procentriole formation in multiciliated cells.

4) An interesting notion is that PLK4, STIL and the PLK4 scaffold CEP152/CEP192 are actually not conserved in all organisms that form centrioles. It would be interesting to have a paragraph discussing which organisms have centrosomes but not PLK4, STIL or CEP152/CEP192.

We thank the reviewers for highlighting this important idea. Indeed, several papers and reviews have examined the evolutionary relationships between centrosomes, centriole duplication and the Plk4-STILCep152/Cep192 modules across different organisms. We have expanded on this topic in the Discussion section and included citations to relevant reviews on this topic. Specifically, we point out that Plk4 is indeed missing in various organisms that form and duplicate centrioles, such as *Chlamydomonas* and *Tetrahymena*. Similarly, there are organisms with centrioles that lack STIL or Cep152/Cep192. This suggests that other pathways and mechanisms exist that permit centriologenesis in a Plk4-independent manner, and that certain mammalian cell types may have adapted these mechanisms. We have included the following paragraph:

“Although Plk4 kinase function was needed for canonical centriole duplication in proliferating basal cells, it was dispensable for post-mitotic centriole amplification. […] Thus, even though components of the centriole assembly machinery are generally conserved throughout evolution, our results suggest that certain mammalian cell types may have adapted mechanisms to initiate centriologenesis independent of Plk4 kinase function.”